# *HFE* Genotype, Ferritin Levels and Transferrin Saturation in Patients with Suspected Hereditary Hemochromatosis

**DOI:** 10.3390/genes12081162

**Published:** 2021-07-28

**Authors:** Miriam Sandnes, Marta Vorland, Rune J. Ulvik, Håkon Reikvam

**Affiliations:** 1Department of Clinical Science, University of Bergen, N-5021 Bergen, Norway; miriam.sandnes@uib.no (M.S.); rune.ulvik@uib.no (R.J.U.); 2Department of Cancer Genomics, Haukeland University Hospital, N-5021 Bergen, Norway; marta.vorland@helse-bergen.no; 3Department of Medicine, Haukeland University Hospital, N-5021 Bergen, Norway

**Keywords:** iron, hemochromatosis, ferritin, transferrin saturation

## Abstract

*HFE* hemochromatosis is characterized by increased iron absorption and iron overload due to variants of the iron-regulating *HFE* gene. Overt disease is mainly associated with homozygosity for the C282Y variant, although the H63D variant in compound heterozygosity with C282Y (C282Y/H63D) contributes to disease manifestation. In this observational study, we describe the association between biochemical findings, age, gender and *HFE* genotype in patients referred from general practice to a tertiary care referral center for diagnostic workup based on suspected hemochromatosis due to persistent hyperferritinemia and *HFE* variants. C282Y and H63D homozygosity were, respectively, the most and least prevalent genotypes and we found a considerable variation in transferrin saturation and ferritin levels independent of *HFE* genotype, which may indeed represent a diagnostic challenge in general practice. While our results confirm C282Y homozygosity as the major cause of iron accumulation, non-C282Y homozygotes also displayed mild to moderate hyperferritinemia with median ferritin levels at 500–700 µg/L, well above the reference cut-off. Such findings have traditionally been ignored in the clinic, and initiation of iron depletion has largely been restricted to C282Y homozygotes. Nevertheless, superfluous iron can aggravate pathogenesis in combination with other diseases and risk factors, such as inflammation, cancer and hepatopathy, and this possibility should not be neglected by clinicians.

## 1. Introduction

*HFE* hemochromatosis is an autosomal recessive disorder caused by variants of the iron-regulating *HFE* gene [1]. The phenotypic hallmark involves increased iron absorption, and it represents the most common monogenic disorder in individuals of northern European descent with a prevalence of approximately 1 in 150 to 350.

Of the two most prevalent variants involved in *HFE* hemochromatosis, C282Y and H63D, C282Y homozygosity is the principal defect ultimately generating a sizable parenchymal iron overload associated with increased risk of liver cirrhosis. Accordingly, 60–100% of clinically diagnosed subjects are C282Y homozygotes, although other genotypes may sporadically generate a serious iron overload despite them being considered as low risk [2,3,4]. As a result of incomplete penetrance, there is a great variation in the associated clinical and biochemical phenotypes. Iron-rich food or iron supplements may influence phenotypic penetrance as well, and due to iron loss through pregnancy and menstruation, definite disease manifestation is significantly lower in women than in men [5,6,7].

While reported prevalence of iron-related disease in untreated C282Y homozygotes is generally regarded as low [8], phenotypic penetrance through biochemical perturbations of iron indices is high, with approximately 50% of women and 80% of men developing hyperferritinemia [9]. A progressive increase in ferritin (hyperferritinemia) as a sign of growing iron storages, and an elevated transferrin saturation (Tsat) as a marker of increased iron absorption are the earliest phenotypic abnormalities in *HFE* hemochromatosis [10].

In our region, inhabited by about half a million people, ferritin is the third most requested analysis from primary health care, making up about 80% of all ferritin analyses performed at the laboratory. As a consequence of this screening practice, persistent mild to moderate hyperferritinemia is sporadically detected in patients without clear evidence of disease, but with a hidden genetic risk for developing hemochromatosis. Ferritin is, however, recognized as an acute phase reactant which rises upon numerous conditions unrelated to iron overload. Hyperferritinemia is thus a highly unspecific finding, and the vast majority of patients presenting with elevated ferritin levels do not have hemochromatosis [11].

When reactive hyperferritinemia associated with underlying diseases as well as conditions which may mask hyperferritinemia (e.g., occult bowel bleeding and blood donations) have been thoroughly considered and ruled out, the finding of a coincidental increase in Tsat supports the likelihood of hyperferritinemia being related to an iron overload [5,12,13,14]. When suspicion of iron overload is maintained, patients are often referred to an outpatient hospital clinic or a specialist, for further assessment, follow up and referral for iron depletion through phlebotomy when indicated.

To improve our knowledge of the *HFE* genotype–phenotype, we examined biochemical profiles associated with various *HFE* genotypes in patients referred to our clinic.

## 2. Materials and Methods

### 2.1. Patients

The Regional Ethics Committee (REK Vest) classified the study as quality assurance; thus, we did not need ethical assessment and informed consent. The hospital data protection officer approved the study. For biochemical characterization in relation to the *HFE* variants, we retrospectively reviewed the hospital records of 409 patients who during 2013–2019 were referred from general practice to the outpatient clinic. These patients were, on the basis of clinical assessment, biochemical findings, *HFE* genotype and suspected iron overload, subsequently referred to phlebotomy. The study population encompassed 316 men (77.3%) and 93 women (22.7%) in total.

### 2.2. Methods

Blood samples were analyzed at the hospital’s central laboratory by automatic routine methods validated to meet the highest technical quality demands. *HFE* genotyping was performed at the central laboratory for clinical genetics. Age at referral and laboratory findings from the diagnostic workup were obtained, including the following full blood parameters: hemoglobin (Hb) and white blood cell count (WBC), and the following serum parameters: ferritin, Tsat, C-reactive protein (CRP), alanine transaminase (ALAT) and gamma-glutamyl transferase (GGT). Viral hepatitis was ruled out in all patients through serology.

Based on medical history and laboratory tests, we were able to rule out recent blood donation, pathological bleeding (that might mask hyperferritinemia caused by iron overload), ongoing use of medication such as iron supplements and proton pump inhibitors, significant acute phase reactions (non-specific or due to known acute and chronic disease), toxic, cholestatic or inflammatory liver disease, diabetes mellitus type 2, metabolic syndrome, inflammation, infection, being significantly overweight and superfluous use of alcohol.

Statistical analyses were performed using the GraphPad Prism 9 software (version 9.1.0, San Diego, CA, USA). Groups were compared using two-tailed Mann–Whitney *U*-test. Spearman’s rank correlation coefficient (*r*) was used as a measure of association. Significant *p*-values are indicated in figures with asterisks: (*), (**) and (***), for ≤0.05, ≤0.01 and ≤0.001, respectively.

## 3. Results

### 3.1. Demographic Data and Ferritin Levels at Referral

Median age in men and women was 47.5 and 57.8 years (*p* ≤ 0.001), respectively (Table 1). On the group level, men were younger than women, with, respectively, 58% and 29% being <50 years of age (Table 2). In subgroups covering decades from the age of 30 and onwards, there was a steady, close to linear rise in ferritin with the highest to lowest median across age groups showing a difference of 271 µg/L in men and 276 µg/L in women.

### 3.2. Distribution of HFE Genotypes

The distribution of *HFE* genotypes in our study population is depicted in Figure 1. A total of 139 subjects were homozygous for the C282Y variant (95 men, 44 women) and 88 were compound heterozygous for the C282Y and H63D variants (67 men, 21 women). These two genotypes accounted for 55.5% of the whole study population, corresponding to 51.3% and 69.9% of men and women, respectively. In descending order, the rest was made up of 85 H63D heterozygotes (69 men, 16 women), 71 C282Y heterozygotes (64 men, 7 women) and 26 H63D homozygotes (21 men, 5 women). Accordingly, in both genders, C282Y homozygosity was the most prevalent genotype, while H63D homozygosity was the least prevalent genotype in both genders.

### 3.3. Phenotypic Imprints

Gender-specific phenotypic imprints are related to ferritin levels in Table 3, and to different *HFE* genotypes in Table 4 and Figure 2. Except for C282Y heterozygotes, age at referral was significantly lower for men than for women (Figure 2). Furthermore, male C282Y homozygotes were significantly younger than men with other genotypes, while for women this was only the case when compared to C282Y/H63D compound heterozygotes and H63D heterozygotes. There was no significant difference in age when comparing other genotypes, in neither men nor women (Appendix A).

All subjects had ferritin ≥ 300 µg/L (i.e., above upper reference value for men) except a female 26-year-old C282Y homozygote with ferritin levels of 234 µg/L. Median ferritin was significantly higher in men than women (*p* ≤ 0.001), although the highest value of 4645 µg/L was found in 45-year-old women. Among men, the highest ferritin value was observed in a 62-year-old exhibiting a ferritin level of 2536 µg/L (Table 3).

Of men, 59.2% had ferritin 501–900 µg/L, and 27.5% ≥ 900 µg/L. Corresponding levels for women were 55.9% and 11.9%. At all ferritin levels between 300–2000 µg/L, median age was higher among women than men, irrespective of *HFE* genotype (Table 3). Furthermore, in men median age was kept stable, just below 50 years at increasing ferritin levels from 501 to 1200 µg/L, and in women around 66 years with ferritin between 701 and 1200 µg/L.

Up to ferritin 900 µg/L, male median Tsat was rather stable, around 42–43%, whereafter it increased linearly to 72.4% as ferritin stepwise increased towards 2000 µg/L. Similarly, apart from an unexplainable drop in the median of five individuals, median Tsat in women was practically linear from 43.5% to 66.6% at ferritin levels up to 1200 µg/L.

In men, there was no real change in median ALAT, which was kept around 41–46 U/L, well below the upper reference limit of 70 U/L, at any ferritin level up to 2000 µg/L. A similar trend was seen in women up to a ferritin level of 900 µg/L. Above this level, ALAT was markedly increased in single subjects.

C282Y homozygous men and women showed the highest median ferritin of 855 and 641 µg/L, with 91.6% and 68.2% of the values being ≥ 500 µg/L, respectively (Table 4). In men with other genotypes, median ferritin varied between 625 and 711 µg/L, with genotypic proportions of ferritin ≥ 500 µg/L ranging from 70.1% to 95.2%. In total, 86.7% of male subjects had ferritin ≥ 500 µg/L. The corresponding findings in female non-C282Y homozygotes were median ferritin between 495 and 640 µg/L, and genotypic proportions with ferritin ≥ 500 µg/L ranging from 40.0% to 85.7%. A total of 67.7% of the female study population had ferritin ≥ 500 µg/L.

Except for C282Y homozygotes, where median ferritin was significantly higher in men than women (*p* ≤ 0.001), there was no significant gender difference in ferritin levels in relation to genotypes (Figure 2).

Median Tsat was a slightly, although insignificantly, higher among women than men (46.7% vs. 44.1%). When evaluating separate *HFE* genotypes, male Tsat tended to surpass that of the female group with a weak statistical significance only for C282Y/H63D compound heterozygotes (Figure 2). Median Tsat above the threshold, defined as the upper male reference level of 45.0%, was, with one negligible exception in male C282Y/H63D compound heterozygotes, only found in C282Y homozygotes with a median of 70.0% in men and 68.3% in women (Table 4). In total, 89.5% of these men and 88.6% of these women had Tsat ≥ 45%. The median Tsat of the male group was 44.1% with 49.4% above the threshold, and in women the corresponding values were 46.7% and 54.8%, respectively (Table 4). In men, C282Y homozygotes showed a significantly higher median Tsat compared to all genotypes (Table 4, Appendix A). This was also the case in C282Y homozygote women (median 68.3%), except for H63D homozygotes (median 41%) (*p* = 0.051), which only accounted for five subjects in total.

On a group level, median ALAT and GGT levels were significantly higher in men than women, although they were well within the reference range for both genders (Table 1). When related to genotype, however, significant gender differences in ALAT were limited to C282Y homozygotes and H63D heterozygotes (Figure 2). Proportion of subjects with elevated ALAT and GGT levels were too small make inferences about specific genotype associations within the groups (Table 4). As a whole, 13.2% of the men had ALAT > 70 U/L and 6.8% had GGT levels above the gender- and age-specific reference intervals. The corresponding results in women were elevated ALAT and GGT in 18.3% and 10.8% of subjects, respectively. Additionally, there was a weak positive correlation between ferritin and ALAT in both genders: in men, *r* = 0.291 (*p* ≤ 0.001), and in women, *r* = 0.309 (*p* ≤ 0.01) (Appendix A).

### 3.4. Ferritin ≥ 1000 µg/L and Elevated Transferrin Saturation

We defined Tsat levels ≥ 70% as significantly elevated. A total of 53 men (16.8%) and 22 (23.7%) women exhibited such levels, two of which were H63D homozygotes, and the rest being either C282Y homozygous (*n* = 69) or C282Y/H63D compound heterozygous (*n* = 4). Likewise, a pathological ferritin ≥ 1000 µg/L was found in a total of 61 subjects (14.9% of total population), which included 17.4% of the male population (*n* = 55) and 6.5% of the female population (*n* = 6) (Table 5). The majority of these men (52.7%) and half of these women were C282Y homozygotes.

In women with ferritin ≥ 1000 µg/L, median CRP, ALAT and GGT were significantly higher compared to women with ferritin < 1000 µg/L. Among men, a similar difference was only seen for ALAT (Figure 3).

A total of 11.4% (*n* = 36) of the total male population and 5.4% (*n* = 5) of the total female population presented with a combined ferritin ≥ 1000 µg/L and Tsat ≥ 45%. Of these subjects, 70% (26 men, 3 women) were C282Y homozygotes (Table 5). There was no significant correlation between ferritin and age, CRP, Tsat, ALAT or GGT levels for these subjects.

### 3.5. CRP and Ferritin Correlation

A significant difference in CRP levels between men and women was detected (Table 1), although median levels were 1.0 and 2.0 mg/L, respectively, and well below the upper reference limit of 5 mg/L. Nevertheless, 10% of men and 22% of women had increased CRP of up to 15–19 mg/L, which indicated inflammation with a potential confounding effect that could contribute to the observed hyperferritinemia. There was in fact a weak positive correlation between CRP and ferritin levels in men (*r* = 0.205, *p* ≤ 0.001) and women (*r* = 0.236, *p* ≤ 0.05) (Appendix A). There was, however, no significant difference in ferritin levels when comparing subjects with and without abnormal CRP, neither in men (*p* = 0.081), nor in women (*p* = 0.293). The ferritin–CRP correlation was strongest in C282Y heterozygous men (*r* = 0.399, *p* ≤ 0.01) and in H63D heterozygous women (*r* = 0.545, *p* ≤ 0.05).

## 4. Discussion

In this observational study, we describe the association between biochemical findings, age, gender and *HFE* genotype in patients referred from general practice to a tertiary care referral center for diagnostic workup on the basis of persistent hyperferritinemia and suspected hereditary hemochromatosis. Only patients with *HFE* variants were included (Figure 1). Comorbidities contributing to a reactive hyperferritinemia were thoroughly investigated, but could not be completely ruled out. Thus, even if the median CRP was normal, some patients had a small increase, and a weak positive correlation between CRP and ferritin levels was detected (Appendix A). A low-grade subclinical inflammation could therefore not be completely excluded, however, with a negligible impact, if any at all.

For example, non-alcoholic fatty liver disease may contribute to reactive, inflammatory hyperferritinemia, and it may also contribute to hepatic iron overload itself through what is referred to as dysmetabolic iron overload syndrome [15]. The wide dispersion of ALAT and GGT values as shown in Table 3 might indicate such an association. While some patients were referred to hepatic magnetic resonance imaging or liver elastography to evaluate the iron burden of the liver, this was not routine practice.

In the preclinical stage, when a convincing correlation between elevated Tsat and ferritin may be absent, the initial diagnostic workup of suspected hemochromatosis often proves to be difficult, not least because hyperferritinemia is a non-specific marker of pathological iron overload [16]. With Tsat being an important marker for genetic penetrance in *HFE* hemochromatosis and a measure of iron accumulation rate, our results confirm C282Y homozygosity as a superior cause of abnormally increased iron accumulation at the population level of this specific area. This complies with the current opinion that the genetic strength required to generate a sizable iron overload, which is big enough to cause clinical disease, is largely restricted to C282Y homozygosity [2,17]. This is valid for both genders; however, women are hit several years after men (Figure 2).

Of all subjects, 85% did not comply with the most important hallmark for complete penetrance of *HFE* hemochromatosis, which is Tsat within the range of 70–100% [18]. However, when evaluating separate genotypes, a convincing hemochromatotic biochemical pattern emerged in C282Y homozygote men with a median ferritin which was significantly higher compared to the other *HFE* genotypes (Table 4, Appendix A), except H63D homozygotes. While median Tsat in C282Y homozygote women equaled that in men, ferritin was not significantly increased beyond that of non-C282Y homozygotes, with a median around 600 µg/L (Table 4, Appendix A).

Not more than 27% of C282Y homozygote men and 7% of women (at postmenopausal age) had a serious risk profile for developing liver complications, which is generally acknowledged as Tsat ≥ 45% and ferritin > 1000 µg/L (Table 5) [19,20,21]. Even though this finding is in line with the fact that C282Y homozygotes are not obligatorily exposed to the development of a dangerous iron overload, one must realize that they were relatively young with a median age of 40 years, and many would therefore be at increasing risk in the years to come. A similar risk profile was found in ten men and two women with other *HFE* genotypes.

An early sign of damage is leakage of ALAT from affected liver cells [13,22]. In accordance with this, serum concentration of ALAT was significantly higher in patients with ferritin ≥ 1000 µg/L compared to those with levels < 1000 µg/L, although median ALAT did not surpass the upper reference limit, in contrast to women, who showed a moderate increase (Figure 3).

As shown in Figure 2, the non-C282Y homozygotes displayed mild to moderate hyperferritinemia, up to 500–700 µg/L, which may be explained by increased iron storage, but external or environmental impact, and not to mention more rare genetic conditions, may also be causative. However, median Tsat in different genotypes was 32–46% and a consistent Tsat of above 26% and 23% was found in *HFE* wildtype men and women, respectively, in a Norwegian hemochromatosis screening which comprised around 65,000 volunteers [23]. In addition, it should be noted that 32% of the men and 24% of the women had Tsat > 45 % (Table 3). Confounding effects due to extreme iron supplementation or regular intake of iron-rich food, such as red meat, was not a reality in our study population.

Nevertheless, we have shown that there was a minimal difference between Tsat in the various non-C282Y homozygote genotypes, the maximum being 9% in women and 13% in men. This was also indirectly suggested by the corresponding small difference between the highest and lowest median ferritin values, which was not more than 86 µg/L in men and 145 µg/L in women. The slight variation in iron overload can be attributed to individual factors such as age, lifestyle and environmental impact.

Historically, non-C282Y homozygote subjects with a stable, mild to moderate iron overload with hyperferritinemia < 1000 µg/L have been considered to be at low risk of serious disease [24]. However, the issue is controversial and has been revitalized in light of recently growing evidence that superfluous iron certainly may aggravate the pathogenesis of other, contemporary diseases such as inflammation, cancer, alcoholism and hepatopathy [25,26,27,28,29,30]. This possibility should not be neglected by clinicians.

Notwithstanding that superfluous iron is preferably deposited in the liver parenchyma, other parenchymatous tissues may be targeted as well, and malfunction might be expressed as (mild) diffuse clinical symptoms or an increase in disease-specific symptoms. Even if this is difficult to prove, the potential pathogenetic impact of iron has gained substantial interest and has been suggested in a wide spectrum of diseases other than liver disease. Of particular interest, is the hypothetical role of iron in degenerative neurological diseases, such as Alzheimer’s and Parkinson’s disease [31,32].

As a whole, the hemochromatotic phenotype indeed depends on the susceptibility of the molecular changes induced by the genetic variants to “external” factors, not to mention other rare variants of which an unknown number awaits discovery. To translate the described associations of this study into rationalized, individual patient care is indeed challenged by the great range and variation in Tsat and ferritin levels in all *HFE* genotypes (Figure 2). Extreme individual Tsat, combined with the highly elevated ferritin values detected in non-C282Y homozygotes (Table 5), may be a consequence of such hitherto unknown confounding factors. Novel genetic variants and digenic models of inheritance have been identified [33,34,35], although much is still unknown regarding environmental and genetic cofactors contributing to iron overload. The modifying effects of epigenetics, miRNA and other gene-regulating mechanisms may also help explain the rather substantially variable phenotype which at present is observed, but only partly understood.

## 5. Conclusions

The considerable variation in Tsat and ferritin, independent of *HFE* genotype, may indeed represent a diagnostic challenge in general practice. This is in particular true for ferritin, which on a regular basis is the diagnostic entrance to abnormal iron metabolism and suspicion of hemochromatosis. On the other hand, permanently increased Tsat > 45% is a rather specific marker for increased iron absorption and genetic penetrance in *HFE* hemochromatosis.

## Figures and Tables

**Figure 1 genes-12-01162-f001:**
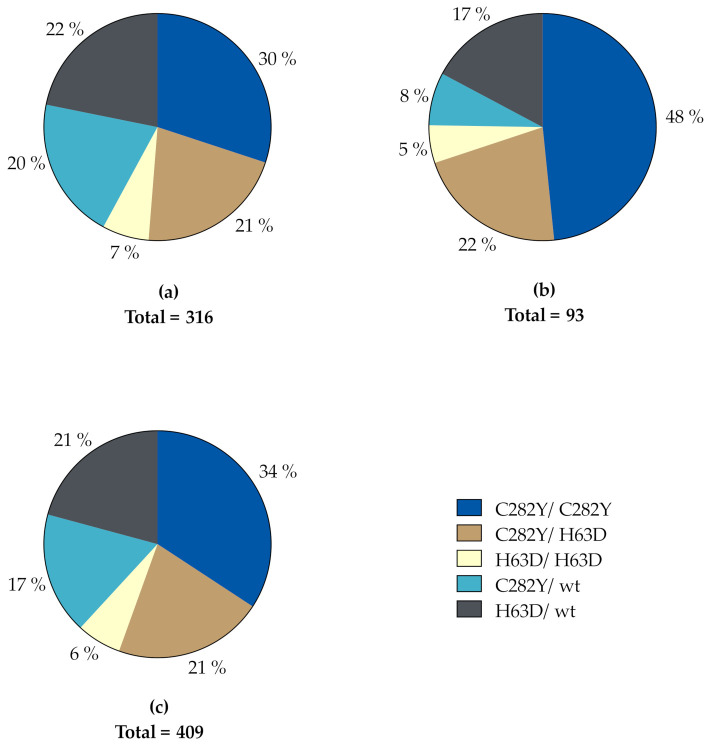
Distribution of *HFE* genotypes in (**a**) men, (**b**) women and (**c**) the total study population. wt, wildtype.

**Figure 2 genes-12-01162-f002:**
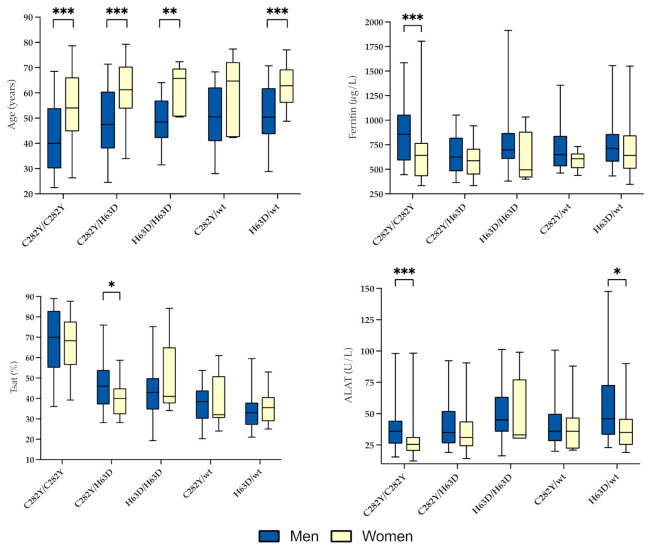
Genotypic differences in age, ferritin, transferrin saturation (Tsat) and alanine aminotransferase (ALAT) levels by gender. Boxes represent the 25th and 75th percentile and whiskers represent the 5th and 95th percentile. Significant *p*-values are indicated in figures with asterisks: * ≤0.05; ** ≤0.01; *** ≤0.001.

**Figure 3 genes-12-01162-f003:**
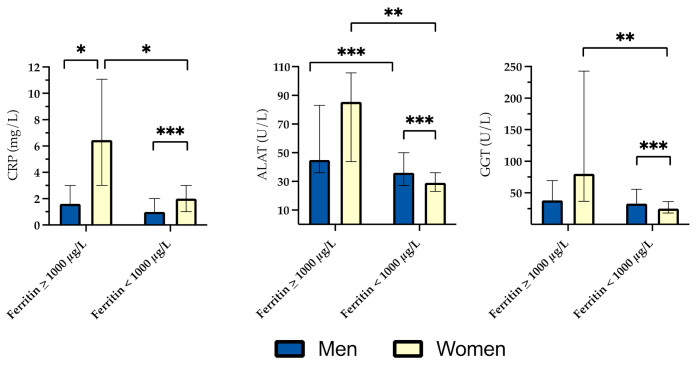
Median C-reactive protein (CRP), alanine aminotransferase (ALAT) and gamma-glutamyl transferase (GGT) in subjects with and without significantly elevated ferritin. Bars represent interquartile range. Significant *p*-values are indicated in figures with asterisks: * ≤0.05; ** ≤0.01; *** ≤0.001.

**Table 1 genes-12-01162-t001:** Descriptive statistics for patients included in the study.

	Ref. Range ^†^		Observations	Median	Range	*p*-Value
Age (years)	N/A	Men	316	47.5	16.3–85.2	≤0.001
		Women	93	57.8	25.1–79.7	
Hb (g/dL)	13.4–17.0	Men	341	16.0	13.6–25.7	≤0.001
	11.7–15.3	Women	97	14.4	12.7–16.9	
WBC (10^9^/L)	3.5–11.0	Men	240	6.2	3.6–12.1	0.876
		Women	73	6.3	3.4–10.5	
CRP (mg/L)	<5	Men	306	1.0	0.1–19	≤0.001
		Women	91	2.0	0.3–15	
Tsat (%)	15–45	Men	316	44.1	11.9–93.0	0.119
	10–40	Women	93	46.7	24.0–89.0	
Ferritin (µg/L)	34–300	Men	316	712	310–2536	≤0.001
	18–240	Women	93	631	234–4645	
ALAT (U/L)	10–70	Men	310	39.0	11–243	≤0.001
	10–45	Women	93	29.0	11–126	
GGT (U/L)	10–80 (15–115 ^¥^)	Men	307	34.0	9–346	≤0.001
	10–45 (10–75 ^¥^)	Women	93	27.0	12–403	

Hb, hemoglobin; WBC, leukocyte particle concentration; CRP, C–reactive protein; Tsat, transferrin saturation; ALAT, alanine aminotransferase; GGT, gamma-glutamyl transferase. ^†^ at Haukeland University Hospital. ^¥^ men and women ≥ 40 years of age.

**Table 2 genes-12-01162-t002:** Ferritin levels at age split in decades.

Age (Years)	Men		Ferritin (µg/L)		Women		Ferritin (µg/L)	
	*n*	%	Median	Range	*n*	%	Median	Range
<30	40	12.6	582	310–2078	5	5.4	644	234–764
30–39	60	19.0	645	355–1637	4	4.3	425	330–648
40–49	84	26.6	700	349–1734	18	19.4	485	328–4645
50–59	59	18.7	818	418–2000	22	23.6	633	346–1696
60–69	58	18.4	723	375–2536	28	30.1	650	386–1841
>70	15	4.7	853	446–1381	16	17.2	701	359–1153
Total	316	100	712	310–2536	93	100	631	234–4645

**Table 3 genes-12-01162-t003:** Median (range) age and biochemical indices at increasing ferritin levels.

**Men**								
**Ferritin (µg/L)**	***n***	**%**	**Age (yrs)**	**Hb (g/dL)**	**Tsat (%)**	**CRP (mg/L)**	**ALAT (U/L)**	**GGT (U/L)**
309–500	42	13.3	39.4 (16.4–70.8)	16.0 (14.4–17.3)	43.0 (20–80)	1.0 (0.2–11)	34.0 (11–103)	26.0 (9–168)
501–700	113	35.8	47.6 (19.4–76.8)	15.8 (13.6–18.9)	42.0 (11.9–87)	1.0 (0.1–12)	34.5 (13–148)	30.0 (12–214)
701–900	74	23.4	48.8 (22.1–74.1)	16.0 (14.1–25.7)	42.5 (22–89)	1.0 (0.4–19)	41.0 (14–153)	35.0 (16–346)
901–1000	32	10.1	47.0 (27.4–73.1)	16.2 (13.6–19.1)	48.8 (23–93)	2.0 (0.9–13)	42.5 (16–117)	49.0 (11–166)
1001–1200	28	8.9	48.6 (31.3–67.6)	16.0 (13.6–17.8)	51.0 (23–90)	2.0 (0.5–11)	44.0 (24–243)	38.5 (13–233)
1201–1500	12	3.8	57.1 (24.5–85.2)	16.3 (13.8–17.6)	67.3 (31–89)	1.0 (0.2–11)	46.0 (22–117)	30.0 (20–220)
1501–2000	12	3.8	53.5 (36.3–62.5)	16.1 (14.4–17.9)	72.4 (31.6–84)	1.0 (0.5–5)	43.5 (25–179)	45.0 (15–189)
>2000	3	0.9	61.6 (28.6–65.1)	15.4 (14.8–16.8)	59.0 (35.5–91)	6.7 (3–10.3)	144 (83–152)	34.0 (20–183)
**Women**								
**Ferritin (µg/L)**	***n***	**%**	**Age (yrs)**	**Hb (g/dL)**	**Tsat (%)**	**CRP (mg/L)**	**ALAT (U/L)**	**GGT (U/L)**
234–500	30	32.3	52.8 (26.8–74.8)	14.8 (12.9–16.2)	43.5 (24–89)	2.0 (1–15)	26.5 (11–38)	24.0 (12–75)
501–700	32	34.4	57.3 (25.1–79.6)	14.3 (12.7–16.3)	48.2 (25–82.9)	2.0 (0.3–9)	34.0 (14–95)	32.5 (12–219)
701–900	20	21.5	66.3 (28.2–79.7)	14.5 (12.9–16.9)	59.9 (32–86.7)	2.0 (0.9–10)	29.0 (14–104)	23.0 (14–91)
901–1000	5	5.4	66.1 (55.2–72.7)	14.9 (13.5–15.7)	43.0 (28–78)	5.0 (1–6)	40.0 (13–41)	30.0 (13–327)
1001–1200	2	2.2	63.7 (50.5–77.0)	16.0 (15.5–16.4)	66.6 (49–84.2)	7.6 (1.2–14)	94.5 (90–99)	80.0 (70–90)
1501–2000	3	3.2	57.1 (54.7–66.2)	14.4 (14.1–14.4)	46.7 (36–81.0)	4.0 (3.6–10.1)	81.0 (19–126)	41.0 (23–189)
>2000	1	1.1	44.8	14.1	67.6	8.9	52.0	403

yrs, years; Hb, hemoglobin; Tsat, transferrin saturation; CRP, C-reactive protein; ALAT, alanine aminotransferase; GGT, gamma-glutamyl transferase.

**Table 4 genes-12-01162-t004:** Age and biochemical findings in relation to genotype.

		Age	Tsat		Ferritin		CRP ^¶^		ALAT †		GGT ^§^	
Genotype	*n*	Median (yrs)	Median (%)	≥45%	Median (µg/L)	≥500 µg/L	Median (mg/L)	≥5 mg/L	Median (U/L)	Elevated ^a^	Median (U/L)	Elevated ^b^
Men												
C282Y/C282Y	95	40.0	70.0	85 (89.5%)	855	87 (91.6%)	1.0	6 (6.7%)	36.0	8 (8.6%)	27.0	3 (3.3%)
C282Y/H63D	67	47.5	46.0	37 (55.2%)	625	47 (70.1%)	1.4	14 (21.5%)	35.0	6 (9.4%)	38.0	5 (7.8%)
H63D/H63D	21	48.4	43.0	7 (33.3%)	698	20 (95.2%)	1.0	2 (9.5%)	45.0	3 (14.3%)	33.0	1 (4.8%)
C282Y/wt	64	50.5	38.5	15 (23.5%)	649	57 (89.1%)	1.0	6 (9.5%)	36.0	6 (9.5%)	37.5	4 (6.5%)
H63D/wt	69	52.8	33.0	12 (17.4%)	711	63 (91.3%)	1.7	2 (2.9%)	46.0	18 (26.1%)	48.0	8 (11.6%)
Total	316	47.4	44.1	156 (49.4%)	712	274 (86.7%)	1.0	31 (10.1%)	39.0	41 (13.2%)	34.0	20 (6.8%)
Women												
C282Y/C282Y	44	54.0	68.3	39 (88.6%)	641	30 (68.2%)	2.0	8 (18.2%)	25.5	4 (9.1%)	21.0	3 (6.8%)
C282Y/H63D	21	61.3	40.0	5 (23.8%)	587	13 (61.9%)	2.0	4 (20%)	31.0	5 (23.8%)	28.0	3 (14.3%)
H63D/H63D	5	65.7	41.0	2 (40.0%)	495	2 (40.0%)	1.1	1 (25%)	33.0	2 (40.0%)	34.0	0 (0.0%)
C282Y/wt	7	64.7	32.0	2 (28.6%)	607	6 (85.7%)	3.0	2 (28.6%)	36.0	2 (28.6%)	35.0	0 (0.0%)
H63D/wt	16	62.8	35.5	3 (18.8%)	640	12 (75.0%)	2.5	5 (31.3%)	35.0	4 (25.0%)	34.5	4 (25%)
Total	93	57.9	46.7	51 (54.8%)	631	63 (67.7%)	2.0	20 (22.0%)	29.0	17 (18.3%)	27.0	10 (10.8%)

yrs, years; wt, wildtype, Tsat, transferrin saturation; CRP, C-reactive protein; ALAT, alanine aminotransferase; GGT, gamma-glutamyl transferase. ^a^ defined as ALAT > 45 U/L in women and >70 U/L in men. ^b^ defined as GGT > 80 and >115 U/L in men < 40 and ≥ 40 years of age, respectively, and GGT > 45 and >75 U/L in women < 40 and ≥40 years of age, respectively. ^¶^ CRP values are missing for 10 men and 2 women. † ALAT values are missing for 6 men. ^§^ GGT values are missing for 9 men.

**Table 5 genes-12-01162-t005:** Proportion with significantly elevated ferritin and transferrin saturation (Tsat) at referral.

Genotype		Tsat ≥ 70%	Ferritin ≥ 1000 µg/L	Combined Ferritin ≥ 1000 µg/L and Tsat ≥ 45%		
		*n*	*n*	*n*	Age (years)	% of Total
C282Y/C282Y	Men	48	29	26	28.6–73.2 ^†^	27.4
	Women	21	3	3	44.8, 54.7, 66.2	6.8
C282Y/H63D	Men	4	5	3	24.5, 59.6, 85.1	4.5
	Women	0	0	0	−	0
H63D/H63D	Men	1	2	1	55.6	4.8
	Women	1	1	1	50.5	20
C282Y/wt	Men	0	9	3	33.4, 54.2, 62.5	4.7
	Women	0	0	0	−	0
H63D/wt	Men	0	10	3	48.0, 59.1, 61.6	4.3
	Women	0	2	1	77.0	6.3
Total	Men	53	55	36	24.5–85.2 ^†^	11.4
	Women	22	6	5	44.8–77.0 ^†^	5.4

wt, wildtype; Tsat, transferrin saturation. Genotypic population who displayed such characteristics. †, range.

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
