# Peer review of "HFE Genotype, Ferritin Levels and Transferrin Saturation in Patients with Suspected Hereditary Hemochromatosis"

_genes, 2021, doi:10.3390/genes12081162_

Round 1

Reviewer 1 Report

Sandnes et al. describe 409 patients with persistent hyperferritinemia and HFE variants, suspected of hemochromatosis in a retrospective study in a tertiary care referral center. The patients were, based on clinical assessment, biochemical results, HFE genotyping and suspicion of iron overload, referred for phlebotomy treatment.

Of the referred patients 77,3 % were male and 22,7 % female. The median age of the men was 10 years less than that of the women. As for the lab results, the Hb level was significantly higher in men, the CRP was significantly higher in women, although both medians were within the reference range. In men the medians of ferritin, ALAT and GGT were higher than in women, with the medians of ALAT and GGT within the reference range.

With regard to the HFE genotypes: 139 subjects were homozygous for the p.C282Y mutation and 88 were compound heterozygous.

The authors examined biochemical profiles in association with various HFE genotypes. The highest median ferritin was found in p.C282Y homozygous men and women, with higher medians in men.  This gender difference was not found in non-p.C282Y homozygotes. With regard to Tsat and p.C282Y homozygosity a Tsat of ≥ 45 % was seen in 89,5 % of the men and in 88,6 % of the women.

A combination of ferritin ≥ 1000 µg/L and Tsat ≥45 % was found in 36 men and 5 women. Of them 26 respectively 3 were homozygous for p.C282Y.

CRP and ferritin: in 10% of the men and 22 % of the women increased levels of CRP were found, which may indicate inflammation as a possible contribution to hyperferritinemia.

As such the findings are not new and confirm previous publications.

#1 The authors mention possible causes of variation in phenotypes, but they do not point to blood donation and use of medication (proton pump inhibitors).

#2 Hyperferritinemia can be caused by iron overload, but also be the result of an acute phase reaction. Furthermore a.o. the metabolic syndrome and drinking alcoholic beverages may increase ferritin levels. The authors state that secondary hyperferritinemia due to comorbidities was ruled out carefully, but they do not specify which and how.

#3 The authors consider the ferritin levels of these patients as a precise indicator or iron overload, but without information about the history of alcohol intake, BMI, waist circumference and determination of the liver iron concentration by means of MRI of the liver I think this is premature, especially in the non-p.C282Y homozygous subjects. The authors refer to their own paper when they write that: "In the preclinical stage when a convincing correlation between elevated Tsat and ferritin may be absent, the initial diagnostic workup of suspected hemochromatosis often proves to be difficult, not least because hyperferritinemia is a non-specific marker of pathological iron overload". In these cases the MRI LIC may contribute at least in our experience. Especially for the non-p.C282Y homozygous patients the conclusion that there was no doubt about the mild to moderate iron overload is premature. If iron overload is confirmed by MRI further genetic exploration of type 2-4 HH may discover the underlying cause.

Typos:

r 49:  progressively  should be: progressive

r 98: lever should be: level

r 248: obligatory should be: obligatoryly

r 275: has should be: have

General remark: many references are rather old...

Author Response

Reviewer 1

Sandnes et al. describe 409 patients with persistent hyperferritinemia and HFE variants, suspected of hemochromatosis in a retrospective study in a tertiary care referral center. The patients were, based on clinical assessment, biochemical results, HFE genotyping and suspicion of iron overload, referred for phlebotomy treatment.

Of the referred patients 77,3 % were male and 22,7 % female. The median age of the men was 10 years less than that of the women. As for the lab results, the Hb level was significantly higher in men, the CRP was significantly higher in women, although both medians were within the reference range. In men the medians of ferritin, ALAT and GGT were higher than in women, with the medians of ALAT and GGT within the reference range.

With regard to the HFE genotypes: 139 subjects were homozygous for the p.C282Y mutation and 88 were compound heterozygous.

The authors examined biochemical profiles in association with various HFE genotypes. The highest median ferritin was found in p.C282Y homozygous men and women, with higher medians in men.  This gender difference was not found in non-p.C282Y homozygotes. With regard to Tsat and p.C282Y homozygosity a Tsat of ≥ 45 % was seen in 89,5 % of the men and in 88,6 % of the women.

A combination of ferritin ≥ 1000 µg/L and Tsat ≥45 % was found in 36 men and 5 women. Of them 26 respectively 3 were homozygous for p.C282Y.

CRP and ferritin: in 10% of the men and 22 % of the women increased levels of CRP were found, which may indicate inflammation as a possible contribution to hyperferritinemia.

As such the findings are not new and confirm previous publications.

#1 The authors mention possible causes of variation in phenotypes, but they do not point to blood donation and use of medication (proton pump inhibitors).

#2 Hyperferritinemia can be caused by iron overload, but also be the result of an acute phase reaction. Furthermore a.o. the metabolic syndrome and drinking alcoholic beverages may increase ferritin levels. The authors state that secondary hyperferritinemia due to comorbidities was ruled out carefully, but they do not specify which and how.

At first consultation a thorough anamnesis was routinely performed to register acute and chronic disease(s) and lifestyle factors that could influence iron status and cause secondary ferritin increase and variation of transferrin saturation. Based on the history and lab-tests we could rule out blood donation, pathological bleeding (that might mask hyperferritinemi caused by iron overload), use of medication (e.g. iron medication, proton pump inhibitors), significant acute phase reaction (non-specific or due to known acute and chronic disease, liver disease, diabetes mellitus type 2/metabolic syndrome, inflammatory disease, infection, etc), overweight and superfluous use of alcohol.  
This has now been more detailed described in the Methods section (line 89-96).

#3 The authors consider the ferritin levels of these patients as a precise indicator or iron overload, but without information about the history of alcohol intake, BMI, waist circumference and determination of the liver iron concentration by means of MRI of the liver I think this is premature, especially in the non-p.C282Y homozygous subjects. The authors refer to their own paper when they write that: "In the preclinical stage when a convincing correlation between elevated Tsat and ferritin may be absent, the initial diagnostic workup of suspected hemochromatosis often proves to be difficult, not least because hyperferritinemia is a non-specific marker of pathological iron overload". In these cases the MRI LIC may contribute at least in our experience. Especially for the non-p.C282Y homozygous patients the conclusion that there was no doubt about the mild to moderate iron overload is premature. If iron overload is confirmed by MRI further genetic exploration of type 2-4 HH may discover the underlying cause.

Our study was a retrospective investigation of hospital records of patients referred during the foregoing years to the out-patient clinic and reflects the medical routine handling of these patients at the highest specialist level in Norway. The diagnostic procedure aimed at deciding if there was an indication for treatment with phlebotomy. It was not a planned case-control study and therefore advanced diagnostic techniques like MRI was not used routinely. We agree though with the reviewer that in cases of doubt, MRI LIC is very informative, and we use it when necessary.

The reviewer states that: “Especially for the non-p.C282Y homozygous patients the conclusion that there was no doubt about the mild to moderate iron overload is premature”. We agree that the evidence is not completely clear, and therefore we have rewritten this sentence (Line 272-275).

Typos:

r 49:  progressively  should be: progressive

r 98: lever should be: level

r 248: obligatory should be: obligatoryly

r 275: has should be: have

These have been corrected.

Reviewer 2 Report

The authors analyzed associations of different HFE genotypes with ferritin concentration and transferrin saturation, and selected biochemical parameters (ALT, GGTP activities and CRP concentration) in a group of patients who had been sent to tertiary care referral with a clinical suspicion of hereditary hemochromatosis. This clinical suspicion was based on the detection of hyperferritinemia by GP. The issue is very interesting, for many years there has been a discussion on the impact of HFE genotypes, other than C282Y homozygosity (and eventually C282Y/H63D mixed heterozygosity) for the development of iron overload related pathology and in consequence multiple organ damage. Convincing results of such studies are important in everyday clinical practice as they are helpful in making decisions about qualifying for treatment aimed at reducing body iron content.

However, I have critical remarks on the methodology of the work. This paper presents results of an observational study with a a significant limits in the collected and analyzed data.

  1. There is no explanation how authors excluded secondary hyperferritinemia in the recruitment of patients, especially in the context of their own observation: a weak correlation of CRP and ferritin concentrations and presence of 10% of men and 22% of women with increased CRP. The characteristics of the study group are very poor in spite of that the authors have assumed biochemical characteristics in relation to HFE variants. The screening biochemical markers that accompany the excessive iron accumulation of iron in HFE haemochromatosis include blood glucose concentration. Obesity, metabolic (or non alcoholic) fatty liver disease, use of alcohol, HCV infection – these are common reasons of increased ferritin in populations of developed countries. In data presented by authors, the wide dispersion of GGTP activity raises the suspicion that there must have been an additional factor associated with other liver disease in these patients (toxic, cholestatic?).These informations could be added to the supplementary data but the inclusion of these data is necessary in the absence of the control group.
  2. The absence of the control group is the main weakness of the study and makes it impossible to draw firm conclusions. Authors recruited patients who were referred to their center because of hyperferritinemia. So they have an access to data from other subjects who appeared to be free from HFE gene mutations. For better understanding of a real impact of different HFE genotypes (excluding those that make up the defininition of HFE-hemochromatosis) comparison of carriers of different HFE gene variants to carriers of wild-type of the gene would be of great importance, significantly increasing the value of the work.
  3. That is why I do not agree with one of final conclusions that “mild to moderate iron overload was therefore no doubt, effectuated by the genetic trait” . Authors did not present complete data to confirm that it is the genetic trait that is responsible for the abnormalities, not the lifestyle, environmental conditions.

I want to emphasize that I appreciate the value of the examination on the role of HFE variants in the development of iron overload.  

Mistakes to correct: line 98 – group level instead group lever (?)

Table 1 – for hemoglobin: number of men 341 (??) and women 97 (??)

In the section material there is information about “the study population encompassed 316 men 78 (77.3%) and 93 women (22.7%) in total”

Line 101: “In subgroups covering decades from the age of 30 and 99 onwards, there was a steady, close to linear, rise in ferritin with the highest to lowest median showing a difference of 271 in men and 276 μg/L in women. median ferritin” – I suppose that value for women should be 57.

For clinicians-readers who work with such patients, more clear presentation of values with value ranges (not only median) would be more valuable. The wide dispersion of ALT values, GGTP requires explanation – are these only single observations with the high activities??

Author Response

Reviewer 2

The authors analyzed associations of different HFE genotypes with ferritin concentration and transferrin saturation, and selected biochemical parameters (ALT, GGTP activities and CRP concentration) in a group of patients who had been sent to tertiary care referral with a clinical suspicion of hereditary hemochromatosis. This clinical suspicion was based on the detection of hyperferritinemia by GP. The issue is very interesting, for many years there has been a discussion on the impact of HFE genotypes, other than C282Y homozygosity (and eventually C282Y/H63D mixed heterozygosity) for the development of iron overload related pathology and in consequence multiple organ damage. Convincing results of such studies are important in everyday clinical practice as they are helpful in making decisions about qualifying for treatment aimed at reducing body iron content.

However, I have critical remarks on the methodology of the work. This paper presents results of an observational study with a a significant limits in the collected and analyzed data.

  1. There is no explanation how authors excluded secondary hyperferritinemia in the recruitment of patients, especially in the context of their own observation: a weak correlation of CRP and ferritin concentrations and presence of 10% of men and 22% of women with increased CRP. The characteristics of the study group are very poor in spite of that the authors have assumed biochemical characteristics in relation to HFE variants. The screening biochemical markers that accompany the excessive iron accumulation of iron in HFE haemochromatosis include blood glucose concentration. Obesity, metabolic (or non alcoholic) fatty liver disease, use of alcohol, HCV infection – these are common reasons of increased ferritin in populations of developed countries. In data presented by authors, the wide dispersion of GGTP activity raises the suspicion that there must have been an additional factor associated with other liver disease in these patients (toxic, cholestatic?).These informations could be added to the supplementary data but the inclusion of these data is necessary in the absence of the control group.

We agree with these remarks, and have further specified the above-mentioned weaknesses of the study design in the discussion. At first consultation a thorough anamnesis was routinely performed to register acute and chronic disease(s) and life style factors that could influence iron status and cause secondary ferritin increase and variation of transferrin saturation. Based on the history and lab.-tests we could rule out blood donation, pathological bleeding (that might mask hyperferritinemi caused by iron overload), use of medication ( e.g. iron medication, proton pump inhibitors), significant acute phase reaction (non-specific or due to known acute and chronic disease), liver disease, diabetes mellitus type 2/metabolic syndrome, inflammatory disease, infection, overweight and superfluous use of alcohol. We have no explanation of the range of GGT, but toxic damage, cholestatics, or other rare conditions were not suspected. These points have now been detailed in the Methods section (line 89-96).

  1. The absence of the control group is the main weakness of the study and makes it impossible to draw firm conclusions. Authors recruited patients who were referred to their center because of hyperferritinemia. So they have an access to data from other subjects who appeared to be free from HFE gene mutations. For better understanding of a real impact of different HFE genotypes (excluding those that make up the defininition of HFE-hemochromatosis) comparison of carriers of different HFE gene variants to carriers of wild-type of the gene would be of great importance, significantly increasing the value of the work.

Patients were referred by general practitioners with an isolated persistent hyperferritinemia. We agree that inclusion of HFE-wild types as controls, would have added valuable information about the relation of genetics and biochemical laboratory profile. Nevertheless, our study reflects the mutually impact of the different HFE-gene traits on the biochemical phenotype, which by itself is valuable information for clinicians.

  1. That is why I do not agree with one of final conclusions that “mild to moderate iron overload was therefore no doubt, effectuated by the genetic trait” . Authors did not present complete data to confirm that it is the genetic trait that is responsible for the abnormalities, not the lifestyle, environmental conditions.

We agree that the evidence of enlarged iron deposits is not completely clear, and therefore we have rewritten this sentence to: As shown in Figure 2, the non-C282Y homozygotes, displayed mild to moderate hyperferritinemia up to 500 - 700 µg/L” which may be explained by increased iron storage, but also external and not to say more rare conditions may be causative”.

I want to emphasize that I appreciate the value of the examination on the role of HFE variants in the development of iron overload.  

Mistakes to correct: line 98 – group level instead group lever (?)

This has been corrected.

Table 1 – for hemoglobin: number of men 341 (??) and women 97 (??)

In the section material there is information about “the study population encompassed 316 men 78 (77.3%) and 93 women (22.7%) in total”

This has been corrected.

Line 101: “In subgroups covering decades from the age of 30 and 99 onwards, there was a steady, close to linear, rise in ferritin with the highest to lowest median showing a difference of 271 in men and 276 μg/L in women. median ferritin” – I suppose that value for women should be 57.

In this sentence we referred to the difference in median ferritin between the age group with the highest median in women (>70 years vs. 30-39 years). We have further specified this to avoid confusion (Line 109).

For clinicians-readers who work with such patients, more clear presentation of values with value ranges (not only median) would be more valuable. The wide dispersion of ALT values, GGTP requires explanation – are these only single observations with the high activities??

We have edited table 3 and added range for age and biochemical findings across ferritin levels. 

Round 2

Reviewer 1 Report

The authors have answered the questions and explained the way the study has been performed.

The text has been adapted appropriately

Reviewer 2 Report

I accept the reply to my comments. I appreciate that the authors highlighted the weaknesses of their project and introduced several explanations into the text, sections: methodology, results, discussion. Hope, that oncological diseases were also excluded in recruited patients with hyperferritinemia.

Please add this information to the excerpt attached to the methods section.

Still corrections should be done in the text, below find examples of the mistakes

v. 91 - hyperferritinemi

v. 105 - lever

v. 243 - resosnance imaginng, iron burde